# Arrow of Time, Entropy, and Protein Folding: Holistic View on Biochirality

**DOI:** 10.3390/ijms23073687

**Published:** 2022-03-28

**Authors:** Victor V. Dyakin, Vladimir N. Uversky

**Affiliations:** 1Virtual Reality Perception Lab (VRPL), The Nathan S. Kline Institute for Psychiatric Research (NKI), 140 Old Orangeburg Road, Bldg, 35, Orangeburg, NY 10962, USA; 2Department of Molecular Medicine, Byrd Alzheimer’s Research Institute, Morsani College of Medicine, University of South Florida, 12901 Bruce B. Downs Blvd., MDC07, Tampa, FL 33612, USA; vuversky@usf.edu

**Keywords:** time arrow, entropy, protein folding, non-equilibrium, chirality consensus, homochirality, protein aging, non-equilibrium entropy, spontaneous post-translational modifications

## Abstract

Chirality is a universal phenomenon, embracing the space–time domains of non-organic and organic nature. The biological time arrow, evident in the aging of proteins and organisms, should be linked to the prevalent biomolecular chirality. This hypothesis drives our exploration of protein aging, in relation to the biological aging of an organism. Recent advances in the chirality discrimination methods and theoretical considerations of the non-equilibrium thermodynamics clarify the fundamental issues, concerning the biphasic, alternative, and stepwise changes in the conformational entropy associated with protein folding. Living cells represent open, non-equilibrium, self-organizing, and dissipative systems. The non-equilibrium thermodynamics of cell biology are determined by utilizing the energy stored, transferred, and released, via adenosine triphosphate (ATP). At the protein level, the synthesis of a homochiral polypeptide chain of L-amino acids (L-AAs) represents the first state in the evolution of the dynamic non-equilibrium state of the system. At the next step the non-equilibrium state of a protein-centric system is supported and amended by a broad set of posttranslational modifications (PTMs). The enzymatic phosphorylation, being the most abundant and ATP-driven form of PTMs, illustrates the principal significance of the energy-coupling, in maintaining and reshaping the system. However, the physiological functions of phosphorylation are under the permanent risk of being compromised by spontaneous racemization. Therefore, the major distinct steps in protein-centric aging include the biosynthesis of a polypeptide chain, protein folding assisted by the system of PTMs, and age-dependent spontaneous protein racemization and degradation. To the best of our knowledge, we are the first to pay attention to the biphasic, alternative, and stepwise changes in the conformational entropy of protein folding. The broader view on protein folding, including the impact of spontaneous racemization, will help in the goal-oriented experimental design in the field of chiral proteomics.

## 1. Introduction

“Natural selection that favors diverse functional molecules and drives the system to the chirality consensus to attain and maintain high-entropy non-equilibrium states” [1].

Chirality entails the fundamental characteristics of the biological and non-biological worlds, meaning that both complementary domains of the universe obey the same laws of space/time symmetry. Indeed, the quantum stereo-physics of electron and nuclear spins link the prevalent homochirality of biological molecules with fundamental processes, associated with the microscopic domains of the space-time structure. On the other hand, the prevalence of bio-molecular chirality and the bilaterality of cognitive functions represent the mega-domain of space-time symmetry. Contemporary science brings new meaning to the notion, from Kant, that space and time are the universal forms of existence.

Due to the difficulties of bridging fundamental principles with biological subjects, the concept of molecular chirality and its relation to the broad range of physical phenomena are frequently excluded, while considering the physiology and pathology of living organisms. Contrary to such unconscious tradition, we provide, here, a short comprehensive introduction, with the references necessary for a deeper understanding.

It is an axiom that chiral objects exhibit non-trivial interactions with chiral environments. In biology, this statement is transformed to the claim that chiral biomolecules exhibit non-trivial interaction with chiral environments [1,2,3,4,5]. In the almost homochiral human body, enantiomers often exhibit different metabolic bioactivity. This difference is a practical guide for current pharmacology, toxicology, and pharmacokinetics. The advances in the physical sciences reveal that biomolecular chirality, including stereochemistry of proteinogenic AAs, is grounded on the quantum physics of elementary particles. The interactions of massive and massless elementary particles, characterized by the different degrees of handedness (helicity and chirality), are the causal forces for the symmetry breaking in the family of weak vector bosons (both W and Z types have a mass). These so-called ultraweak chiral forces ultimately dictate the symmetry-related behavior to subatomic, atomic, and molecular level processes [4]. Erich Poppitz and Yanwen Shang name the complex of theoretical models, describing the symmetry-related behavior as the “theory space” [5]. This complex includes Gauge Group Theories, String Theory, Standard Model of particle physics, and Quantum Gravity Theory. The spatial distribution of electron density (i.e., the symmetry of atomic orbitals), for each of the atoms in the Mendeleev table, is determined by the complex chirality of all elementary particles. According to Noether’s principle, the change in the energy of an atomic and molecular system is usually associated with an alteration to the symmetry and vice versa. The illustrative examples are hydrogen and carbon atoms (Figure 1a–c) [6]. Carbon is of particular interest as the chiral center of the proteinogenic AAs. Carbon has four half-filled sp3 hybrid orbitals at the ground state, which form a chirogenic tetrahedron (Figure 1b). The symmetry of atomic and molecular orbitals is a function of energy level, susceptible to the asymmetry of the environment. The most studied examples of spin isomers are molecular hydrogen (H_2_) and molecular oxygen (O_2_), which exhibit sharply different reactivity in the ground and excited states associated with the different symmetry of molecular orbitals. The spatial symmetry of atomic orbitals is a primary determinant of AAs’ chirality, and consequently, protein folding entropy.

Chiral substitutions of AAs residues have drastic consequences for peptide structure, solubility, assembly, aggregation, and function [7,8]. The evolutionary selection of protein sequence, structure, stability, and function, driven by the hidden power of entropy, involves the mechanism of AAs substitution [7,8]. As a result, we have a limited number of proteinogenic AAs in the human organism, with differential frequency of appearance in the proteome [9]. The significance of stereospecific biochemical reactions is not restricted to proteinogenic AAs. A well-known example of non-proteinogenic biologically essential L-AAs is L-DOPA, a precursor to several neurotransmitters (dopamine, adrenaline, and noradrenaline). However, we will focus primarily on the role of thermodynamics of proteinogenic AAs. Proteinogenic amino acids (AAs) are organic molecules that contain carbon C (C_6_), hydrogen (H_1_), nitrogen (N_7_), oxygen (O_8_), and sulfur (S_16_) atoms (CHNOS) (see Table 1) [10].

Although the configurations of electron clouds of all CHNOS atoms contribute to the AAs chirality, the role of the tetrahedral carbon as a chiral center is the most essential. A distinct role in modulating protein chirality, folding, and function, belongs to phosphorus (P_15_), participating in the most abundant PTMs—phosphorylation. The overlapping of electron orbitals constitutes the inter-atomic forces, providing an opportunity to form non-organic and organic (such as proteins) molecular structures in different states of aggregation. The chirality of any molecular structure (including AAs, proteins, fatty acids, lipids, and DNA/RNA) is determined by the 3D electron density distribution. From this perspective, it is not surprising that spin-dependent electron transport in chiral molecules was observed experimentally [11,12,13,14].

The coupling of electron-nuclear spins symmetry to the molecular level of chirality is a critical element for understanding the stereo-specific mechanisms of two competing forms (enzymatic and spontaneous) of post-translational modifications (PTMs) in proteins (PTMs Enz and PTM Sp). PTMs Enz are controlled by the catalytic activity of specific enzymes and create various out-of-equilibrium products, which are prone to spontaneous condensation and racemization, accompanied by the non-trivial behavior of entropy. The significance of PTMs Sp, occurring due to the presence of L-AA side chains, is highlighted by the fact that racemization interferes with the many other essential forms of biochemical reactions, including oxidation (cysteine, histidine, methionine, phenylalanine, tryptophan, and tyrosine), intra- and inter-residue cyclization (asparagine, aspartic acid, glutamic acid, glutamine, and N-terminal diketopiperazine (DKP) formation), and β-elimination reactions (cysteine, cystine, serine, and threonine).

## 2. Space-Time Symmetry and Entropy

Spice-time symmetry and entropy are fundamental features of Nature that encapsulate both the non-biological and biological worlds. The first and second laws of thermodynamics, complemented by Noether’s Theorem, operating with the concepts of space, time and entropy, allow a unifying, non-contradicting interpretation of a wide variety of phenomena, ranging from Black Hole irradiation [15,16] to photon chirality [17,18], protein synthesis [19], and even to the evolution of brain function [20]. For more information regarding space-time integrity and the link between the entropy and time direction, see [21,22,23,24]. Following the idea of Schrödinger, pointing to the non-trivial behavior of entropy in the living system [25], the interdisciplinary discussion of the non-equilibrium (NE) thermodynamics of biological systems revealed the inherent link between the concept of entropy, chirality, and spontaneous racemization, linked to the evolution of life, and biological aging (Figure 1d) [26,27,28]. Even the entropic brain hypothesis recently emerged [29,30,31]. Unfortunately, however, the prevalent molecular chirality was not always taken into account in these considerations. The significance of entropy transformation at the protein level assumes that there are no cellular functions, including cell division, migration, and proliferation, which can be understood without consideration of the molecular chirality. Consequently, the fundamental significance of biochirality at the cellular level assumes that features of the organism/brain morphology, behavior, and cognition cannot be understood without considering the molecular and cellular chirality. However, until recently, there were no studies systematically describing the entropy production rate associated with irreversible biological processes [32,33]. Two postulates of the non-biological and biological information processing theories state that an increasing entropy decreases the information in the system, and decreasing entropy increases the system’s information. Accordingly, the information is the reduction in entropy (i.e., uncertainty) in the receiver [33]. The fidelity of information flow from DNA to RNA to proteins to cells, which constitutes the basis for living organisms, relies on the chirality of their molecular constituents [2,3]. It is not trivial that there is less information content in DNA than in the proteins assembled based on genetic code [34]. Due to advances in the appreciation of biochirality, the intuitive perception of the link between bio-molecular chirality and brain laterality and asymmetry of cognitive function is gradually replaced by a system of experimental facts [35,36,37,38,39]. The evolution of biological symmetry is evident in the animal proteome, cell signaling, and body/brain morphology, originating, mainly, due to the changes in the gene regulatory networks (GRNs) [39,40,41]. The human genome contains ∼25,000 protein-coding genes, which are differentially expressed, in response to the various intracellular and extracellular cues. Current theories explain biological symmetry as a pattern, determined, mainly, by the combination of phylogenetic and ontogenetic constraints.

## 3. Biological Language of Protein Conformation

### 3.1. Protein Homochirality

Among 20 evolutionarily selected proteinogenic amino acids (AAs), 19 exist in two stereoisomeric forms of L- and D-amino acids (L-AAs and D-AAs) [42]. It was long assumed that D-AAs are uncommon in mammals, until a small amount of free D-serine (D-Ser) was found in the mammalian brain (serving as a neurotransmitter) and many other parts of human body (Figure 1d) [43]). Maintaining a low D-AA/L-AAs ratio is a vital biological function because excessive free and protein-bound D- AAs are toxic for many cellular functions. The observation of a preferred transport of D-Ser and D-Pro across the blood-brain barrier [44] confirms the significance of low D-AAs levels for proper brain function. The chirality of AAs has a strong impact on the conformational dynamics and the stability of proteins. Proteins containing homochiral L-AAs typically (with a rare exception) consist of right-handed (R-handed) a-helices [45,46], whereas left-handed (L-handed) helices are relatively rare in nature. The thermodynamic stability of helices, and other secondary structure elements, is different for peptides containing only L-AAs, only D-AAs, and mixed (L- and D-) AAs. The stability of an L-AAs-based helix is low in the water and higher in the membrane lipid environment. Based on these experimentally derived facts, we can expect that time-dependent spontaneous racemization will have a significant impact on the conformation, stability, dynamics and physiological function of proteins.

Ribosomal protein synthesis operates only with L-AAs. Three primary sources of D-AAs in the body are enzymatic racemization, diet, gut microbiota [47], and spontaneous racemization. The physiology of the first three mechanisms is widely stated, while the thermodynamically favored (i.e., characterized by high probability) spontaneous racemization’s contribution and consequences are mostly overlooked. Ribosomal protein synthesis results in the NE pool of homochiral proteins.

Consequently, although entropy of the newly synthesized AAs sequence is dramatically reduced, the possibility of entropy increase remains, due to the homochirality of AAs [48].

The biological language of protein symmetry is further based on the crosstalk between PTMs^Enz^ and PTM^Sp^. The language of biological information processing is the evolving target of multi-disciplinary fields of science, grounded in the basic concepts of NE thermodynamics: homochirality, entropy, spontaneity, and irreversibility [49]. The irreversible increase in entropy is traditionally recognized as the time arrow of physicochemical processes of non-biological and biological nature [34,49,50,51,52,53]. The nature of time arrows influences the direction of the biological evolution [42], age-dependent decline in adaptation capacity [43], and the lifespan of an organism [43,51,52,53]. Proteins, synthesized exclusively by the ribosomes, are modified and degraded by various enzymatic and spontaneous pathways. Spontaneous racemization can be enhanced by the network of PTMs^Enz^ (such as phosphorylation) and a broad spectrum of PTMs^Sp^, such as deamination, isomerization, and carbamylation, all contributing to the molecular aging and to the aging of the organism [54,55,56]. The most frequently occurring PTMs^Enz^ are protein phosphorylation and dephosphorylation, which provide the switch between active and inactive states of proteins. The interplay between enzymatic PTMs (e.g., phosphorylation) and spontaneous racemization has been shown to be a principal determinant of the age-associated pathogenic pathways [57]. There is now general agreement that the long-lived proteins (LLPs) are inherently vulnerable to various time-dependent non-enzymatic modifications, including unfolding, misfolding, damage, and pathological aggregation [58,59]. A broader view on the integrative physiology of biopolymers in the organism also points to the coupling between enzymatic phosphorylation and racemization reactions. The homochirality-concerning interaction of biomolecules, such as DNAs, RNAs, proteins, and lipids, remains a long-standing scientific puzzle [58,60,61]. The final product–proteins produce multi-pathways feedback, which influences DNA functions. One of the most studied players in protein–DNA interaction is the enzyme RNA Polymerase (RNAP). Functionally significant phosphorylation of the C-terminal domain (CTD) occurs at the Ser, Tyr, and Thr residues, which are also prone to spontaneous racemization [61].

In this short review, we summarize what is known, regarding the role of entropy in homochiral proteins synthesis, as well as protein functions and the age-dependent degradation of protein homochirality.

### 3.2. Fundamental Roots of Homochirality

Most advanced researchers consider protein homochirality as an inevitable consequence of the universal fundamental physical processes in the quantum realm of elementary particles, atomic structures, dynamics of chemical reactions, living organisms, as well as their immediate eco-environment, and in the extraterrestrial scale of events [62]. An alternative view (more archaic) is that biochemistry plays a more important role than abiotic chemical or physical processes, thereby assuming that biomolecular homochirality is a consequence of life, rather than a prerequisite for life.

Evolutionary-selected homochirality of proteins synthesis allows for minimization in the amount of information required for the explicit coding of AAs sequences by the nucleic acids [63,64]. The homochiral AAs chain undergoes enzymatic and spontaneous modifications, accompanied by a variety of phase transitions (PhTs) [25,28,65,66]. PTMs^Enz^ and PTMs^Sp^ are greatly increasing the complexity of the proteome, promoting the flow of biological information. L-AAs chains of newly synthesized homochiral proteins experience the branched sequence of stereo-processing, and, therefore, afford the opportunity of self-organization [67]. The hierarchical organization of a protein structure, such as the secondary, tertiary, and quaternary structures, is predetermined by the initial homochirality. At the second level of the hierarchy (secondary structures), the α-helix usually has a right-handed conformation. However, the survey of the abundance of left-handed helices (L-helixes) in the Protein Data Bank (PDB) revealed 31 L-helices in a set of 7284 verified proteins [45]. The initial chirality can be switched several times from right to left, at the following hierarchical levels [68]. The commonality of this phenomenon to all proteins remains to be verified. Notably, the complex of spontaneous phase transitions (PhTs^Sp^) is protein-specific (i.e., it is unique for any polypeptide chain) and is highly stereoselective. A major part of PTMs^Enz^ and PTMs^Sp^ was evolutionarily selected as the pathway to enlarge the functional conformational space of the native state (NS). Another part of PTMs^Sp^ is known to disrupt the natural course of molecular events, often leading to pathogenic misfolding, malfunction, and aggregation of proteins. One of the most studied forms of pathogenic reactions is racemization [69,70]. As an unavoidable consequence of pathogenic PTMs, we can expect that even small (limited only to one or two AAs) racemization can trigger the deviation from the NS. The functional NS of protein has relatively low entropy because its conformational freedom is highly restricted [71]. Spontaneous racemization of any particular AAs residues may occur at any step in the chain of PTMs. However, it has remained unknown which stage is the most critical for triggering pathology.

### 3.3. Spontaneous Racemization

The homochirality and racemization of proteins are two opposite biochemical processes. Newborn organisms maintain the homochirality of protein synthesis and utilize a small number of D-AAs for most vital physiological processes. Protein folding pathways are prone to malfunction, driven by a pathogenic spontaneous stereoselective reaction, accompanied by aberrant liquid-to-solid phase transitions (PhTs), leading to stepwise, age-associated misfolding, including fibrillization and aggregation. During a lifetime, spontaneous racemization is an inevitable factor, contributing to proteins, cells, and organism aging [57]. Spontaneous racemization frequently coexists with the abnormal PTM^Enz^, especially phosphorylation. This is not surprising, because phosphorus (P) is known as a chiro-genic atom. The 3-D structure of electron orbitals for the P atom plays a causal role in the chirality of many phosphorus ions and molecular structures, including phosphoranes and adenosine phosphates [72,73,74,75]. Adenosine phosphates (APs) are essential, biologically relevant chiral molecules, which define the energy balance of most living systems [76]. They exist in various forms, such as ATP, ADP, and AMP, and their interconversion phenomena are directly related to multiple stereo-specific processes occurring in the living cell [77].

At the time of death, the ratio of D/L is slightly shifted toward an increasing D-isoform, due to relative or partial racemization. After death, AAs continue to racemize at a rate dependent on the particular AA, the temperature, pH, radiation, and the chemical environment. Postmortem racemization reactions are slow on the lifetime scale and rapid on the terrestrial geologic time scale (AAs are completely racemized (D/L = 1.0 in <5–10 million years) [62]. The fundamental significance of protein homochirality assumes no less of a fundamental role for racemization.

## 4. Entropy

“The loss of conformational entropy is a major contribution in the thermodynamics of protein folding” [78].

Statistical consideration of entropy, based on the Boltzmann equation, reveals non-trivial entropy-driven ordering, in out-of-equilibrium soft matter systems, such as colloids, hard sphere suspensions, and liquid crystals [34,79]. An accurate theoretical description of protein-folding dynamics can be achieved based on the consideration of both the Boltzmann–Planck and Bose-Einstein-Planck distribution of conformational states [80]. Corresponding entropy-driven disorder–order and order–disorder PhTs^Sp^ were experimentally observed in the biological [81] and non-biological [79] colloidal systems.

From a thermodynamics perspective, biological self-organization is an entropy reduction process, inherent for the NE homochiral systems, that can be observed at the molecular, cellular, and morphological levels [34]. Our primary concern is focused on the entropy behavior during the distinct states of protein configuration (Figure 2), including the spontaneous racemization of NS. Both the native and unfolded states are highly heterogeneous. Spontaneous self-assembly (folding) of most homochiral proteins into NS occurs at a very short time interval (in the order of 10^−6^ to 10^−1^ s) [82,83]. Rapid folding to NS prevents the impact of unwonted spontaneous racemization, which is, generally, a much slower process. For example, the conformation time for L-Asp 58 to d-isoAsp58 in αA-crystallin is compared to the organism’s lifetime [84].

It has been shown that the polymerization of free (dispersed in the cytosol) L-AAs during ribosome-driven protein biosynthesis results in a considerable entropy reduction [85]. However, the degree of freedom for the newly synthesized, mostly unfolded, polypeptide chain remains relatively high (Figure 2, State II). A relatively high entropy level of this newly synthesizing chain changes over time, due to two major processes: entropy reduction due to the step-vise polypeptide condensation (spatial localization) into native globular conformation and entropy production by the step-vise racemization of AAs [86].

Experimental studies of stepwise folding reveal that α-helices have lower entropy, on average, than β-sheets, and entropy levels of both types of ordered secondary structures are lower than that of coil/irregular/loop regions, and all of them are lower than the entropy of the unfolded chain [87]. The NS of homochiral proteins are trapped in an energy minimum. This state is, thermodynamically, only relatively stable and, therefore, is prone to various spontaneous relaxations, leading to the formation of many aberrant conformations and aggregative states. Such spontaneous processes can be triggered by the racemization of AAs [70,88]. Spontaneous racemization is known as one of the unavoidable degrading forces of the functional state of proteins. There are multiple states, which are thermodynamically more stable than native protein forms. These are amyloids, fibrils, and other types of aggregates, which are associated with protein aging, translated to the cellular and organism levels. However, despite recent advances in the understanding of protein folding and mis-folding, researchers mostly ignore the homochirality of the NS and, correspondently, overlook the contribution of spontaneous racemization to protein aging [89].

In terms of the quantitative thermodynamic variables, spontaneous protein folding from the initial mostly unfolded chain of AAs (state I) to the NS (state II) is accompanied by the decrease in the Gibbs free energy (D G) and the decrease in the entropy of the system (D S^Fold^), due to the formation of a more ordered state.
D G = D H − T * D S(1)

From the thermodynamic perspective, some other processes should compensate for the loss of D S^Fold^. One of these processes is the gain of enthalpy due to the binding accompanied by condensation and polymerization. Two others are the increase in the entropy of solvents and increase in the entropy of proteins due to the loss of protein homochirality (racemization) [25]. The complexity of PTMs is a thermodynamically driven mechanism to convert the epigenetic signals (stressors) into biological responses. The chain of interlinked molecular, cellular, systemic, and organism-level biological events allow consideration of non-enzymatic racemization, which is the set of the time-dependent and irreversible PTMs^Sp^, as the fundamental multivariable determinant of protein aging and biological aging of an organism as a whole. An organism’s aging and its counterpart spontaneous molecular events are inevitable processes. However, the body of evidence suggests that epigenetic aging could be, if not preventable, then treatable, for specific stereo-selective therapeutic interventions [89,90,91,92,93,94].

The ribosome-mediated translational machinery (TM) of the cell is capable of the incorporation of exclusively L-AAs in the primary polypeptide chain of synthesized proteins [95]. This newly synthesized chain is mostly unfolded and exists as a highly dynamic conformational ensemble, containing many different conformations. In fact, because of the very large number of degrees of freedom in an unfolded polypeptide chain, the molecule has an astronomical number of possible configurations (polypeptide chain of 100 AAs can adopt more than 5 × 10^47^ different shapes [71]).

Furthermore, these highly conformational dynamics are attributed to the high level of entropy in the system, which is reduced at each step of protein folding. The essential advantages attributed to protein homochirality are clearly articulated [63,64,96,97].

A homochiral chain of L-AAs (characterized by a high level of entropy) is more prone to spontaneous condensations and secondary structure formation (such as α-helices and β-sheets) than the heterochiral or racemic variants. The chain of spontaneous condensations lowers the number of accessible states to the folded protein, reducing the translational, configurational, rotational, and vibrational components of entropy. Protein homeostasis depends on energy-consuming processes.

Consequently, de novo protein synthesis requires ATP hydrolysis for the formation of peptide bonds of the initial polypeptide chain (state II). Next, this newly synthesized chain undergoes a series of PhTs^Sp^, eventually leading to the formation of the NS of protein (state III). The transition from state I to state II (NS) is accompanied by a reduction in the configurational entropy of a polypeptide chain, accompanied by the increase in the solvent (cytosol or lipid environment) [98]. One should also keep in mind that the resulting NS is a complex NE system, with limited stability, that can be disrupted by many pathogenic conditions, such as ATP exhaustion [99] and racemization [70]. The transition from NS to aberrant pathogenic states is believed to occur through partially unfolded conformation [100]. The increased flexibility of protein sub-units in such conformations promotes several spontaneous PTMs, including racemization [101,102]. The possibility of spontaneous racemization of NSs agrees with the effect of the AA’s chirality, propagated through the peptide backbone observed in the model peptides [101]. Under these conditions, the NS of protein spontaneously loses its functions, reverting to the dynamic ensemble of the non-native and non-functional conformations. Among such non-native denaturalizing conformations are those generated by the partial racemization of protein-bound AAs, incorporated into the polypeptide chain of a protein. In both cases, the nascent homochiral polypeptide chain and its corresponding racemic variant are characterized by the high level of entropy, relative to the functional NS. The critical difference is that the newly synthesized polypeptide chain is in the NE state, which will eventually fold into the NS with lower entropy, which, over time, will undergo spontaneous racemization, thus, leading to partial unfolding and aggregation, whereas the racemic variant typically represents an equilibrium form.

## 5. Epilogue

Due to the cumulative contributions of physicists and mathematicians, such as Nikolai Lobachevsky (1792–1856), Hendrik Antoon Lorentz (1853–1928), Jules Henri Poincaré (1854–1912), Max Karl Ernst Ludwig Plank (1858–1947), Hermann Minkowski (1864–1909), Marie Salomea Skłodowska Curie (1867–1934), Albert Einstein (1879–1955), Amalie Emmy Noether (1882–1935), Erwin Rudolf Josef Alexander Schrödinger (1887–1861), Wolfgang Ernst Pauli (1900–1958), Werner Karl Heisenberg (1901–1976), and Paul Adrien Maurice Dirac (1902–1984), the concept of space experienced a dramatic transformation, from the classical Euclidean to the modern space-time continuum, reflecting the significance of the chirality concept [103,104,105]. In parallel, the enormous progress in our understanding of the significance of molecular chirality was made by Jean-Baptiste Biot (1774–1862), Louis Pasteur (1822–1895), Joseph Achille Le Bel (1847–1930), Jacobus Henricus van’t Hoff (1852–1911), Hermann Emil Louis Fisher (1852–1919), and many others. As a result, it becomes clear that chirality, entropy, and energy are entangled parameters of the physical and chemical objects driving their spatial behavior, playing particularly important roles in protein folding and structural dynamics. The entropy of the bio-molecular system can be partitioned into energy-dependent components: translational, rotational, orientational, and vibrational [105]. It is notable that PhTs between enzyme functional states are usually connected to conformational changes, involving electron or proton transport (spin transport) and directional shifts of a group of atoms. These microscopic fluxes and stereospecific conformations result in entropy production, driven by non-equilibrium concentrations of substrates and products (see Entropy production principle [106,107,108].

It is a common belief that studying the molecular mechanism of protein folding, from the original (unfolded) homochiral polypeptide chain to native conformation, has represented one of the biggest challenges in biochemistry and molecular biology [109,110,111,112,113]. In this field, the most intense debates were and remain focused on the role of entropy in protein self-assembly, into nano and mesoscale structures [107,108,109]. Experimental observation of the effects of electron-nuclear-photon spins coupling allows for the determination of the 3-D shape of the protein structure [110]. Resolving the protein structure, in turn, promotes the understanding of cell and organism physiology. The fact that the heat-induced racemization of free AAs is ten times slower than AAs included in the polypeptide chain [114,115] explains the vulnerability of LLPs, to spontaneous racemization during the lifetime of an organism. In agreement with the thermodynamic laws, the entirety of available results allows authors to articulate the summary axiom (Dyakin-Uversky axiom): the non-equilibrium state of the homochiral ensemble of proteins allows the bidirectional (non-monotonic, decreasing–increasing) behavior of entropy.

Although new achievements are commonly recognized, the adequate understanding of the link of chirality and entropy phenomena in biology, which embrace very remote areas, such as the electron nuclear spins coupling, protein folding, metabolism, memory, cognition and consciousness, are just in an embryonic state [116,117,118,119,120,121].

## Figures and Tables

**Figure 1 ijms-23-03687-f001:**
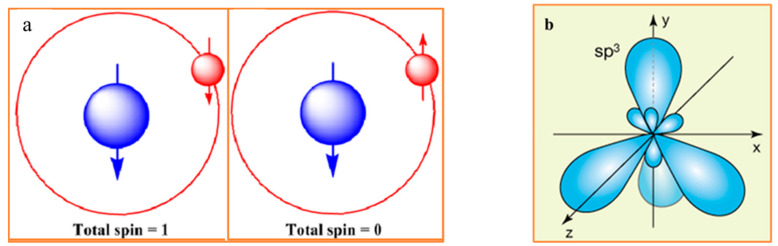
(**a**) In a hydrogen atom, aligning the spins of the proton (blue) and the electron (red) increases the atom’s total spin from zero to one. Antiparallel electron and proton spin alignments (**left**), the system occurs in a lower energy state. Parallel electron and proton spins configuration (**right**) has slightly higher energy state. Adapted from [6]. (**b**) Carbon atom has four equivalent half-filled sp3 hybrid orbitals at the ground state, which form a chirogenic tetrahedron. (**c**) Three major determinants of spontaneous racemization in the non-equilibrium bio-molecular system: thermodynamics, entropy, and chirality. (**d**) The regions of human body containing trackable levels of D-AAs. 1. Brain. 2. Endocrine tissue. 3. Lung. 4. Blood and immune system. 5. Liver. 6. Male Tissue. 7. Adipose tissue. 8. Heart and skeletal muscle. 9. Gastrointestinal tract. 10. Pancreas. 11. Kidney. 12. Female tissue. 13. Skin.

**Figure 2 ijms-23-03687-f002:**
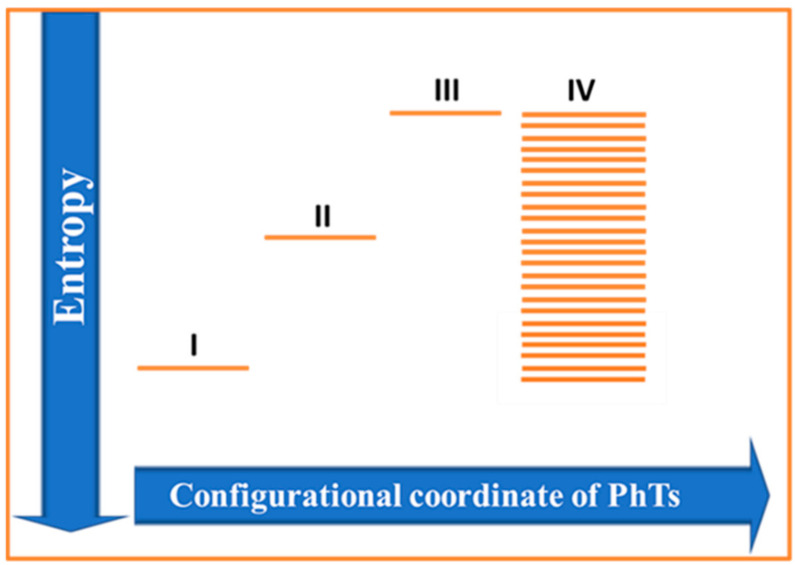
Schematic representation of entropy-driven protein self-assembly. From a statistical perspective, racemization is associated with increase in entropy, whereas protein folding is accompanied by the entropy decrease. Distinct states of protein configuration. I—Free AAs (cytosol). II—Unfolder chain of AAs. III—Native state (NS) of ordered globular protein. IV—The set of discrete states reflects the different degrees of spontaneous racemization and illustrates the complexity of the resulting outputs of such racemization. Importantly, spontaneous racemization and spontaneous protein folding are associated with Gibbs’s free energy decrease. The sequences of phase transitions I-II-III and III-IV are both related to the decrease in the Gibbs free energy but the opposite behavior of entropy. Although the discrete stepwise entropy reduction due to protein condensation (folding) is well-known, the increase in entropy due to the spontaneous racemization is mostly overlooked. Arrows on the right side of the diagram schematically represent degrees of the racemization one would expect during the lifetime of an organism (metabolically active state) and spontaneous complete postmortem racemization (metabolically inactive state).

**Table 1 ijms-23-03687-t001:** Atomic content of the proteinogenic AAs (oxygen (O), carbon (C), nitrogen (N) sulfur(S) and hydrogen (H) atoms) and their frequency distribution in protein sequences found in the Swiss-Prot. database (Swiss-ProtFraction). Adopted with modification from [10].

Proteino-Genic Amino Acids (AAs)
	Amino Acid	H(1)	C(6)	N(7)	O(8)	S(16)	Swiss-PrtFr
1	Alanine (Ala/A)	7	3	1	2	0	0.0777
2	Arginine (Arg/R)	15	6	4	2	0	0.0526
3	Asparagine (Asp/N)	8	4	2	3	0	0.0437
4	Aspartate (Asp/D)	6	4	1	4	0	0.053
5	Cysteine (Cys/C)	7	3	1	2	1	0.0157
6	Glutamate (Glu/E)	8	5	1	4	0	0.0692
7	Glutamine (Glx/Z)	10	5	2	3	0	0.0532
8	Glycine (Gly/G)	5	2	1	2	0	0.0691
9	Histidine (His/H)	10	6	3	2	0	0.0227
10	Isoleucine (Ile/I)	13	6	1	2	0	0.0591
11	Leucine (Leu/L)	13	6	1	2	0	0.096
12	Lysine (Lys/K)	15	6	2	2	0	0.0595
13	Methionine (Me/Mt)	11	5	1	2	1	0.0238
14	Phenylaninr (Phe/F)	11	9	1	2	0	0.0405
15	Proline (Pro/P)	10	5	1	2	0	0.0469
16	Serine (Ser/S)	7	3	1	3	0	0.0694
17	Threonine (Thr/T)	9	4	1	3	0	0.055
18	Tryptoohan(Trp/W)	11	11	2	2	0	0.0118
19	Tyrosin (Tyr/Y)	11	9	1	3	0	0.0311
20	Valine (Val/V)	11	5	1	2	0	0.0667
Non-Proteino-genic amino acids (AAs)
1	L-Carnitine

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
