# Peer review of "Arrow of Time, Entropy, and Protein Folding: Holistic View on Biochirality"

_ijms, 2022, doi:10.3390/ijms23073687_

Round 1
Reviewer 1 Report
In this review “Arrow of Time, Entropy, and Protein Folding: Holistic View on Biochirality”, the authors broadly discuss the importance of chirality in protein folding, aging and degradation. They have very systematically reported excerpts taken from research articles published elsewhere, on the concepts of entropy, protein chirality and spontaneous racemization. Overall, the review is written and presented legibly to convincingly highlight the link of chirality and entropy in biological systems. However, in the absence of a focussed research on this topic this review remains far from being truly complete.
I have a few minor concerns regarding the review that needs to addressed in order to improve its readability for wider scientific community.
- The introduction is unorganized and mixed up without a clear direction. Please rewrite to encompass the essence of what the review is going to be.
- There are numerous typographical errors in the review that should be carefully checked and corrected.
Author Response
Dear Rev. One.
- Thank for you suggestions and recommendations.
- Introduction is modified according to your request (lines 129-140).
- Typographical error ate fixed (see track changes).
Thank very much.
Victor V. Dyakin.
Reviewer 2 Report
Dear authors,
This review covers an interesting subject.
However, the authors just bring together some basic concepts without having a real red line to connect all these.
After careful consideration of the manuscript, one of the conclusions that can be drawn is that this article must be carefully revised again. In the case of references, there are multiple typographical errors that need to be remedied.
For this reason, at this level the article must be carefully analyzed
Author Response
Dear Rev. Two
- Thank for you suggestions and recommendations.
- According to your suggestions the text was revised.
See the changes in the line 129-140; 335-339; 444-451.
- New references are provided [114-115; 453, 455}
Thank very much.
Victor V. Dyakin.
Round 2
Reviewer 2 Report
Dear authors,
The MS was modified according with reviewers comments.
Please read carefully again your MS and improve the quality of this review.
There are still some issues that deserves to be investigated.
a. Please check Table 1 (line 117) please use L-carnitine instead of L-canitine
b. The phrase from 337-339 is almost identical with 337-339!
c. Many references need to be converted into appropriate format (18,27,37,45,60,74,77,78 - no authors, 85, 86, 97)
I hope that all these observation will help the authors to submit a publishable version of this review.
Kind regards
Author Response
Dear reviewer two
All suggested corrections are done, including:
- Table 1 (line 117) L-canitine is replaced by L-carnitine.
- The phrase repetition from 300 -303 and 337-339 is fixed.
- The references (18,27,37,41, 45,60,74, 76 77,78, 85, 86, and 97) are converted into appropriate format.
Thanks very much for improvement the quality of the text.
Victor V. Dyakin
Round 3
Reviewer 2 Report
Dear authors,
The last version is an improved one.
Good luck